# TRUTH OR BACKPROPAGANDA? AN EMPIRICAL INVESTIGATION OF DEEP LEARNING THEORY

**Micah Goldblum**[*]
Department of Mathematics
University of Maryland
goldblum@umd.edu

**Jonas Geiping**[*]
Department of Computer Science and Electrical Engineering
University of Siegen
jonas.geiping@uni-siegen.de

**Avi Schwarzschild**
Department of Mathematics
University of Maryland
avi1@umd.edu

**Michael Moeller**
Department of Computer Science and Electrical Engineering
University of Siegen
michael.moeller@uni-siegen.de

**Tom Goldstein**
Department of Computer Science
University of Maryland
tomg@umd.edu

## ABSTRACT

We empirically evaluate common assumptions about neural networks that are widely held by practitioners and theorists alike. In this work, we: (1) prove the widespread existence of suboptimal local minima in the loss landscape of neural networks, and we use our theory to find examples; (2) show that small-norm parameters are not optimal for generalization; (3) demonstrate that ResNets do not conform to wide-network theories, such as the neural tangent kernel, and that the interaction between skip connections and batch normalization plays a role; (4) find that rank does not correlate with generalization or robustness in a practical setting.

## 1 INTRODUCTION

Modern deep learning methods are descendent from such long-studied fields as statistical learning, optimization, and signal processing, all of which were built on mathematically rigorous foundations. In statistical learning, principled kernel methods have vastly improved the performance of SVMs and PCA (Suykens & Vandewalle, 1999; Schölkopf et al., 1997), and boosting theory has enabled weak learners to generate strong classifiers (Schapire, 1990). Optimizers in deep learning are borrowed from the field of convex optimization , where momentum optimizers (Nesterov, 1983) and conjugate gradient methods provably solve ill-conditioned problems with high efficiency (Hestenes & Stiefel, 1952). Deep learning harnesses foundational tools from these mature parent fields.

Despite its rigorous roots, deep learning has driven a wedge between theory and practice. Recent theoretical work has certainly made impressive strides towards understanding optimization and generalization in neural networks. But doing so has required researchers to make strong assumptions and study restricted model classes.

In this paper, we seek to understand whether deep learning theories accurately capture the behaviors and network properties that make realistic deep networks work. Following a line of previous work, such as Swirszcz et al. (2016), Zhang et al. (2016), Balduzzi et al. (2017) and Santurkar et al. (2018), we put the assumptions and conclusions of deep learning theory to the test using experiments with both toy networks and realistic ones. We focus on the following important theoretical issues:

---

[*]Authors contributed equally.

- Local minima: Numerous theoretical works argue that all local minima of neural loss functions are globally optimal or that all local minima are nearly optimal. In practice, we find highly suboptimal local minima in realistic neural loss functions, and we discuss reasons why suboptimal local minima exist in the loss surfaces of deep neural networks in general.

- Weight decay and parameter norms: Research inspired by Tikhonov regularization suggests that low-norm minima generalize better, and for many, this is an intuitive justification for simple regularizers like weight decay. Yet for neural networks, it is not at all clear which form of $\ell_2$-regularization is optimal. We show this by constructing a simple alternative: biasing solutions toward a non-zero norm still works and can even measurably improve performance for modern architectures.

- Neural tangent kernels and the wide-network limit: We investigate theoretical results concerning neural tangent kernels of realistic architectures. While stochastic sampling of the tangent kernels suggests that theoretical results on tangent kernels of multi-layer networks may apply to some multi-layer networks and basic convolutional architectures, the predictions from theory do not hold for practical networks, and the trend even reverses for ResNet architectures. We show that the combination of skip connections and batch normalization is critical for this trend in ResNets.

- Rank: Generalization theory has provided guarantees for the performance of low-rank networks. However, we find that regularization which encourages high-rank weight matrices often outperforms that which promotes low-rank matrices. This indicates that low-rank structure is not a significant force behind generalization in practical networks. We further investigate the adversarial robustness of low-rank networks, which are thought to be more resilient to attack, and we find empirically that their robustness is often lower than the baseline or even a purposefully constructed high-rank network.

## 2 LOCAL MINIMA IN LOSS LANDSCAPES: DO SUBOPTIMAL MINIMA EXIST?

It is generally accepted that "in practice, poor local minima are rarely a problem with large networks." (LeCun et al., 2015). However, exact theoretical guarantees for this statement are elusive. Various theoretical studies of local minima have investigated spin-glass models (Choromanska et al., 2014), deep linear models (Laurent & Brecht, 2018; Kawaguchi, 2016), parallel subnetworks (Haeffele & Vidal, 2017), and dense fully connected models (Nguyen et al., 2018) and have shown that either all local minima are global or all have a small optimality gap. The apparent scarcity of poor local minima has lead practitioners to develop the intuition that bad local minima ("bad" meaning high loss value and suboptimal training performance) are practically non-existent.

To further muddy the waters, some theoretical works prove the *existence* of local minima. Such results exist for simple fully connected architectures (Swirszcz et al., 2016), single-layer networks (Liang et al., 2018; Yun et al., 2018), and two-layer ReLU networks (Safran & Shamir, 2017). For example, (Yun et al., 2019) show that local minima exist in single-layer networks with univariate output and unique datapoints. The crucial idea here is that all neurons are activated for all datapoints at the suboptimal local minima. Unfortunately, these existing analyses of neural loss landscapes require strong assumptions (e.g. random training data, linear activation functions, fully connected layers, or extremely wide network widths) — so strong, in fact, that it is reasonable to question whether these results have any bearing on practical neural networks or describe the underlying cause of good optimization performance in real-world settings.

In this section, we investigate the existence of suboptimal local minima from a theoretical perspective and an empirical one. If suboptimal local minima exist, they are certainly hard to find by standard methods (otherwise training would not work). Thus, we present simple theoretical results that inform us on how to construct non-trivial suboptimal local minima, concretely generalizing previous constructions, such as those by (Yun et al., 2019). Using experimental methods inspired by theory, we easily find suboptimal local minima in the loss landscapes of a range of classifiers.

Trivial local minima are easy to find in ReLU networks – consider the case where bias values are sufficiently low so that the ReLUs are "dead" (i.e. inputs to ReLUs are strictly negative). Such a point is trivially a local minimum. Below, we make a more subtle observation that multilayer perceptrons (MLPs) must have non-trivial local minima, provided there exists a linear classifier that

performs worse than the neural network (an assumption that holds for virtually any standard benchmark problem). Specifically, we show that MLP loss functions contain local minima where they behave identically to a linear classifier on the same data.

We now define a family of low-rank linear functions which represent an MLP. Let "rank-$s$ affine function" denote an operator of the form $G(\mathbf{x}) = A\mathbf{x} + \mathbf{b}$ with $\text{rank}(A) = s$.

**Definition 2.1.** Consider a family of functions, $\{F_\phi : \mathbb{R}^m \to \mathbb{R}^n\}_{\phi \in \mathbb{R}^P}$ parameterized by $\phi$. We say this family has *rank-$s$ affine expression* if for all rank-$s$ affine functions $G : \mathbb{R}^m \to \mathbb{R}^n$ and finite subsets $\Omega \subset \mathbb{R}^m$, there exists $\phi$ with $F_\phi(\mathbf{x}) = G(\mathbf{x})$, $\forall \mathbf{x} \in \Omega$. If $s = \min(n, m)$ we say that this family has *full affine expression*.

We investigate a family of L-layer MLPs with ReLU activation functions, $\{F_\phi : \mathbb{R}^m \to \mathbb{R}^n\}_{\phi \in \Phi}$, and parameter vectors $\phi$, i.e., $\phi = (A_1, \mathbf{b}_1, A_2, \mathbf{b}_2, \ldots, A_L, \mathbf{b}_L)$, $F_\phi(\mathbf{x}) = H_L(f(H_{L-1}...f(H_1(\mathbf{x}))))$, where $f$ denotes the ReLU activation function and $H_i(\mathbf{z}) = A_i\mathbf{z} + \mathbf{b}_i$. Let $A_i \in \mathbb{R}^{n_i \times n_{i-1}}$, $\mathbf{b}_i \in \mathbb{R}^{n_i}$ with $n_0 = m$ and $n_L = n$.

**Lemma 1.** *Consider a family of L-layer multilayer perceptrons with ReLU activations $\{F_\phi : \mathbb{R}^m \to \mathbb{R}^n\}_{\phi \in \Phi}$, and let $s = \min_i n_i$ be the minimum layer width. Such a family has rank-$s$ affine expression.*

*Proof.* The idea of the proof is to use the singular value decomposition of any rank-$s$ affine function to construct the MLP layers and pick a bias large enough for all activations to remain positive. See Appendix A.1. □

The ability of MLPs to represent linear networks allows us to derive a theorem which implies that arbitrarily deep MLPs have local minima at which the performance of the underlying model on the training data is equal to that of a (potentially low-rank) linear model. In other words, neural networks inherit the local minima of elementary linear models.

**Theorem 1.** *Consider a training set, $\{(\mathbf{x}_i, y_i)\}_{i=1}^N$, a family $\{F_\phi\}_\phi$ of MLPs with $s = \min_i n_i$ being the smallest width. Consider a parameterized affine function $G_{A,\mathbf{b}}$ solving*

$$\min_{A,\mathbf{b}} \mathcal{L}(G_{A,\mathbf{b}}; \{(\mathbf{x}_i, y_i)\}_{i=1}^N), \qquad \text{subject to } \text{rank}(A) \leq s, \tag{1}$$

*for a continuous loss function $\mathcal{L}$. Then, for each local minimum, $(A', \mathbf{b}')$, of the above training problem, there exists a local minimum, $\phi'$, of the MLP loss $\mathcal{L}(F_\phi; \{(\mathbf{x}_i, y_i)\}_{i=1}^N)$ with the property that $F_{\phi'}(\mathbf{x}_i) = G_{A',\mathbf{b}'}(\mathbf{x}_i)$ for $i = 1, 2, ..., N$.*

*Proof.* See appendix A.2. □

The proof of the above theorem constructs a network in which all activations of all training examples are positive, generalizing previous constructions of this type such as Yun et al. (2019) to more realistic architectures and settings. Another paper has employed a similar construction concurrently to our own work (He et al., 2020). We do expect that the general problem in expressivity occurs every time the support of the activations coincides for all training examples, as the latter reduces the deep network to an affine linear function (on the training set), which relates to the discussion in Balduzzi et al. (2017). We test this hypothesis below by initializing deep networks with biases of high variance.

**Remark 2.1** (CNN and more expressive local minima). Note that the above constructions of Lemma 1 and Theorem 1 are not limited to MLPs and could be extended to convolutional neural networks with suitably restricted linear mappings $G_\phi$ by using the convolution filters to represent identities and using the bias to avoid any negative activations on the training examples. Moreover, shallower MLPs can similarly be embedded into deeper MLPs recursively by replicating the behavior of each linear layer of the shallow MLP with several layers of the deep MLP. Linear classifiers, or even shallow MLPs, often have higher training loss than more expressive networks. Thus, we can use the idea of Theorem 1 to find various suboptimal local minima in the loss landscapes of neural networks. We confirm this with subsequent experiments.

We find that initializing a network at a point that approximately conforms to Theorem 1 is enough to get trapped in a bad local minimum. We verify this by training a linear classifier on CIFAR-10 with

Table 1: Local minima for MLPs generated via various initializations. We show loss, euclidean norm of the gradient vector, and minimum eigenvalue of the Hessian before and after training. We use $500$ iterations of the power method on a shifted Hessian matrix computed on the full dataset to find the minimum eigenvalue. The experiment in the last row is trained with no momentum (NM).

| | At Initialization | | | After training | | |
|---|---|---|---|---|---|---|
| Init. Type | Loss | Grad. | Min. EV | Loss | Grad. | Min. EV |
| Default | 4.5963 | 0.5752 | -1.5549 | 0.0061 | 0.0074 | 0.0007 |
| Lemma 1 | 1.5702 | 0.0992 | 0.03125 | 1.5699 | 0.0414 | 0.0156 |
| Bias+20 | 31.204 | 343.99 | -1.7421 | 2.3301 | 0.0090 | 0.0005 |
| Bias $\in \mathcal{U}(-50, 50)$ | 51.445 | 378.36 | -430.49 | 2.3153 | 0.0048 | 0.0000 |
| Bias $\in \mathcal{U}(-10, 10)$ NM | 12.209 | 42.454 | -47.733 | 0.2198 | 0.0564 | 0.0013 |

weight decay, (which has a test accuracy of $40.53\%$, loss of $1.57$, and gradient norm of $0.00375$ w.r.t to the logistic regression objective). We then initialize a multilayer network as described in Lemma 1 to approximate this linear classifier and recompute these statistics on the full network (see Table 1). When training with this initialization, the gradient norm drops futher, moving parameters even closer to the linear minimizer. The final training result still yields positive activations for the entire training dataset.

Moreover, any isolated local minimum of a linear network results in many local minima of an MLP $F_{\phi'}$, as the weights $\phi'$ constructed in the proof of Theorem 1 can undergo transformations such as scaling, permutation, or even rotation without changing $F_{\phi'}$ as a function during inference, i.e. $F_{\phi'}(\mathbf{x}) = F_{\phi}(\mathbf{x})$ for all $\mathbf{x}$ for an infinite set of parameters $\phi$, as soon as $F$ has at least one hidden layer.

While our first experiment initializes a deep MLP at a local minimum it inherited from a linear one to empirically illustrate our findings of Theorem 1, Table 1 also illustrates that similarly bad local minima are obtained when choosing large biases (third row) and choosing biases with large variance (fourth row) as conjectured above. To significantly reduce the bias, however, and still obtain a sub-par optimum, we need to rerun the experiment with SGD without momentum, as shown in the last row, reflecting common intuition that momentum is helpful to move away from bad local optima.

**Remark 2.2** (Sharpness of sub-optimal local optima). An interesting additional property of minima found using the previously discussed initializations is that they are "sharp". Proponents of the sharp-flat hypothesis for generalization have found that minimizers with poor generalization live in sharp attracting basins with low volume and thus low probability in parameter space (Keskar et al., 2016; Huang et al., 2019), although care has to be taken to correctly measure sharpness (Dinh et al., 2017). Accordingly, we find that the maximum eigenvalue of the Hessian at each suboptimal local minimum is significantly higher than those at near-global minima. For example, the maximum eigenvalue of the initialization by Lemma 1 in Table 1 is estimated as $113,598.85$ after training, whereas that of the default initialization is only around $24.01$. While our analysis has focused on sub-par local optima in training instead of global minima with sub-par generalization, both the scarcity of local optima during normal training and the favorable generalization properties of neural networks seem to correlate with their sharpness.

In light of our finding that neural networks trained with unconventional initialization reach suboptimal local minima, we conclude that poor local minima can readily be found with a poor choice of hyperparameters. Suboptimal minima are less scarce than previously believed, and neural networks avoid these because good initializations and stochastic optimizers have been fine-tuned over time. Fortunately, promising theoretical directions may explain good optimization performance while remaining compatible with empirical observations. The approach followed by Du et al. (2019) analyzes the loss trajectory of SGD, showing that it avoids bad minima. While this work assumes (unrealistically) large network widths, this theoretical direction is compatible with empirical studies, such as Goodfellow et al. (2014), showing that the training trajectory of realistic deep networks does not encounter significant local minima.

## 3   WEIGHT DECAY: ARE SMALL $\ell_2$-NORM SOLUTIONS BETTER?

Classical learning theory advocates regularization for linear models, such as SVM and linear regression. For SVM, $\ell_2$ regularization endows linear classifiers with a wide-margin property (Cortes & Vapnik, 1995), and recent work on neural networks has shown that minimum norm neural network interpolators benefit from over-parametrization (Hastie et al., 2019) . Following the long history of explicit parameter norm regularization for linear models, weight decay is used for training nearly all high performance neural networks (He et al., 2015a; Chollet, 2016; Huang et al., 2017; Sandler et al., 2018).

In combination with weight decay, all of these cutting-edge architectures also employ batch normalization after convolutional layers (Ioffe & Szegedy, 2015). With that in mind, van Laarhoven (2017) shows that the regularizing effect of weight decay is counteracted by batch normalization, which removes the effect of shrinking weight matrices. Zhang et al. (2018) argue that the synergistic interaction between weight decay and batch norm arises because weight decay plays a large role in regulating the effective learning rate of networks, since scaling down the weights of convolutional layers amplifies the effect of each optimization step, effectively increasing the learning rate. Thus, weight decay increases the effective learning rate as the regularizer drags the parameters closer and closer towards the origin. The authors also suggest that data augmentation and carefully chosen learning rate schedules are more powerful than explicit regularizers like weight decay.

Other work echos this sentiment and claims that weight decay and dropout have little effect on performance, especially when using data augmentation (Hernández-García & König, 2018). Hoffer et al. (2018) further study the relationship between weight decay and batch normalization, and they develop normalization with respect to other norms. Shah et al. (2018) instead suggest that minimum norm solutions may not generalize well in the over-parametrized setting.

We find that the difference between performance of standard network architectures with and without weight decay is often statistically significant, even with a high level of data augmentation, for example, horizontal flips and random crops on CIFAR-10 (see Tables 2 and 3). But is weight decay the most effective form of $\ell_2$ regularization? Furthermore, is the positive effect of weight decay because the regularizer promotes small norm solutions? We generalize weight decay by biasing the $\ell_2$ norm of the weight vector towards other values using the following regularizer, which we call *norm-bias*:

$$R_\mu(\phi) = \left| \left( \sum_{i=1}^{P} \phi_i^2 \right) - \mu^2 \right|. \tag{2}$$

$R_0$ is equivalent to weight decay, but we find that we can further improve performance by biasing the weights towards higher norms (see Tables 2 and 3). In our experiments on CIFAR-10 and CIFAR-100, networks are trained using weight decay coefficients from their respective original papers. ResNet-18 and DenseNet are trained with $\mu^2 = 2500$ and norm-bias coefficient 0.005, and MobileNetV2 is trained with $\mu^2 = 5000$ and norm-bias coefficient 0.001. $\mu$ is chosen heuristically by first training a model with weight decay, recording the norm of the resulting parameter vector, and setting $\mu$ to be slightly higher than that norm in order to avoid norm-bias leading to a lower parameter norm than weight decay. While we find that weight decay improves results over a non-regularized baseline for all three models, we also find that models trained with large norm bias (i.e., large $\mu$) outperform models trained with weight decay.

These results lend weight to the argument that explicit parameter norm regularization is in fact useful for training networks, even deep CNNs with batch normalization and data augmentation. However, the fact that norm-biased networks can outperform networks trained with weight decay suggests that any benefits of weight decay are unlikely to originate from the superiority of small-norm solutions.

To further investigate the effect of weight decay and parameter norm on generalization, we also consider models without batch norm. In this case, weight decay directly penalizes the norm of the linear operators inside a network, since there are no batch norm coefficients to compensate for the effect of shrinking weights. Our goal is to determine whether small-norm solutions are superior in this setting where the norm of the parameter vector is more meaningful.

In our first experiment without batch norm, we experience improved performance training an MLP with *norm-bias* (see Table 3). In a state-of-the-art setting, we consider ResNet-20 with Fixup initialization, a ResNet variant that removes batch norm and instead uses a sophisticated initialization

that solves the exploding gradient problem (Zhang et al., 2019). We observe that weight decay substantially improves training over SGD with no explicit regularization — in fact, ResNets with this initialization scheme train quite poorly without explicit regularization and data normalization. Still, we find that *norm-bias* with $\mu^2 = 1000$ and norm-bias coefficient 0.0005 achieves better results than weight decay (see Table 3). This once again refutes the theory that small-norm parameters generalize better and brings into doubt any relationship between classical Tikhonov regularization and weight decay in neural networks. See Appendix A.5 for a discussion concerning the final parameter norms of Fixup networks as well as additional experiments on CIFAR-100, a harder image classification dataset.

Table 2: ResNet-18, DenseNet-40, and MobileNetV2 models trained on non-normalized CIFAR-10 data with various regularizers. Numerical entries are given by $\overline{m}(\pm s)$, where $\overline{m}$ is the average accuracy over 10 runs, and $s$ represents standard error.

| Model | No weight decay (%) | Weight decay (%) | Norm-bias (%) |
|---|---|---|---|
| ResNet | 93.46 ($\pm$0.05) | 94.06 ($\pm$0.07) | **94.86** ($\pm$0.05) |
| DenseNet | 89.26 ($\pm$0.08) | 92.27 ($\pm$0.06) | **92.49** ($\pm$0.06) |
| MobileNetV2 | 92.88 ($\pm$0.06) | 92.88 ($\pm$0.09) | **93.50** ($\pm$0.09) |

Table 3: ResNet-18, DenseNet-40, MobileNetV2, ResNet-20 with Fixup initialization, and a 4-layer multi-layer perceptron (MLP) trained on normalized CIFAR-10 data with various regularizers. Numerical entries are given by $\overline{m}(\pm s)$, where $\overline{m}$ is the average accuracy over 10 runs, and $s$ represents standard error.

| Model | No weight decay (%) | Weight decay (%) | Norm-bias (%) |
|---|---|---|---|
| ResNet | 93.40 ($\pm$0.04) | 94.76 ($\pm$0.03) | **94.99** ($\pm$0.05) |
| DenseNet | 90.78 ($\pm$0.08) | 92.26 ($\pm$0.06) | **92.46** ($\pm$0.04) |
| MobileNetV2 | 92.84 ($\pm$0.05) | **93.64** ($\pm$0.05) | **93.64** ($\pm$0.03) |
| ResNet Fixup | 10.00 ($\pm$0.00) | 91.42 ($\pm$0.04) | **91.55** ($\pm$0.07) |
| MLP | 58.88 ($\pm$0.10) | 58.95 ($\pm$0.07) | **59.13** ($\pm$0.09) |

## 4 KERNEL THEORY AND THE INFINITE-WIDTH LIMIT

In light of the recent surge of works discussing the properties of neural networks in the infinite-width limit, in particular, connections between infinite-width deep neural networks and Gaussian processes, see Lee et al. (2017), several interesting theoretical works have appeared. The wide network limit and Gaussian process interpretations have inspired work on the neural tangent kernel (Jacot et al., 2018), while Lee et al. (2019) and Bietti et al. (2018) have used wide network assumptions to analyze the training dynamics of deep networks. The connection of deep neural networks to kernel-based learning theory seems promising, but how closely do current architectures match the predictions made for simple networks in the large-width limit?

We focus on the Neural Tangent Kernel (NTK), developed in Jacot et al. (2018). Theory dictates that, in the wide-network limit, the neural tangent kernel remains nearly constant as a network trains. Furthermore, neural network training dynamics can be described as gradient descent on a convex functional, provided the NTK remains nearly constant during training (Lee et al., 2019). In this section, we experimentally test the validity of these theoretical assumptions.

Fixing a network architecture, we use $\mathcal{F}$ to denote the function space parametrized by $\phi \in \mathbb{R}^p$. For the mapping $F : \mathbb{R}^P \to \mathcal{F}$, the NTK is defined by

$$\Phi(\phi) = \sum_{p=1}^{P} \partial_{\phi_p} F(\phi) \otimes \partial_{\phi_p} F(\phi), \qquad (3)$$

where the derivatives $\partial_{\phi_p} F(\phi)$ are evaluated at a particular choice of $\phi$ describing a neural network. The NTK can be thought of as a similarity measure between images; given any two images as input, the NTK returns an $n \times n$ matrix, where $n$ is the dimensionality of the feature embedding of the neural network. We sample entries from the NTK by drawing a set of $N$ images $\{x_i\}$ from a dataset,

and computing the entries in the NTK corresponding to all pairs of images in our image set. We do this for a random neural network $f : \mathbb{R}^m \rightarrow \mathbb{R}^n$ and computing the tensor $\Phi(\phi) \in R^{N \times N \times n \times n}$ of all pairwise realizations, restricted to the given data:

$$\Phi(\phi)_{ijkl} = \sum_{p=1}^{P} \partial_{\phi_p} f(\mathbf{x}_i, \phi)_k \cdot \partial_{\phi_p} f(\mathbf{x}_j, \phi)_l \qquad (4)$$

By evaluating Equation 4 using automatic differentiation, we compute slices from the NTK before and after training for a large range of architectures and network widths. We consider image classification on CIFAR-10 and compare a two-layer MLP, a four-layer MLP, a simple 5-layer ConvNet, and a ResNet. We draw 25 random images from CIFAR-10 to sample the NTK before and after training. We measure the change in the NTK by computing the correlation coefficient of the (vectorized) NTK before and after training. We do this for many network widths, and see what happens in the wide network limit. For MLPs we increase the width of the hidden layers, for the ConvNet (6-Layer, Convolutions, ReLU, MaxPooling), we increase the number of convolutional filters, for the ResNet we consider the WideResnet (Zagoruyko & Komodakis, 2016) architecture, where we increase its width parameter. We initialize all models with uniform He initialization as discussed in He et al. (2015b), departing from specific Gaussian initializations in theoretical works to analyze the effects for modern architectures and methodologies.

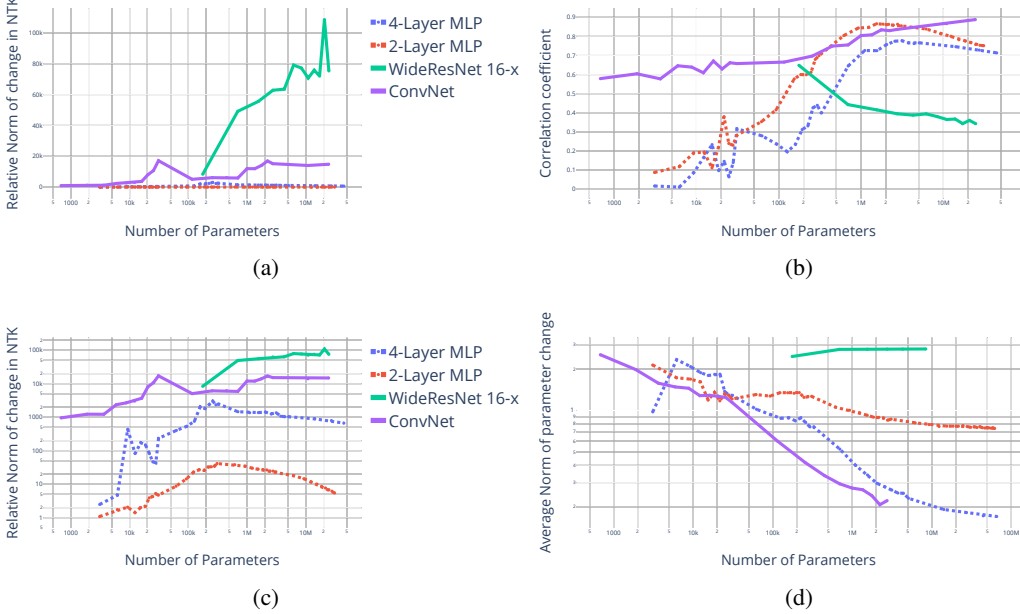

Figure 1: (a) The relative norm of the neural tangent kernel as a function of the number of parameters is shown for several networks. This figure highlights the difference between the behavior of ResNets and other architectures. Figure 1c visualizes the same data in a logarithmic scale. (b) The correlation of the neural tangent kernel before and after training. We expect this coefficient to converge toward 1 in the infinite-width limit for multi-layer networks as in Jacot et al. (2018). We do not observe this trend for ResNets as is clear from the curve corresponding to the WideResNet. (d) The average norm of parameter change decreases for simple architectures but stays nearly constant for the WideResNet.

The results are visualized in Figure 1, where we plot parameters of the NTK for these different architectures, showing how the number of parameters impacts the relative change in the NTK ($||\Phi_1 - \Phi_0||/||\Phi_0||$, where $\Phi_0/\Phi_1$ denotes the sub-sampled NTK before/after training) and correlation coefficient ($\text{Cov}(\Phi_1, \Phi_0)/\sigma(\Phi_1)/\sigma(\Phi_0)$). Jacot et al. (2018) predicts that the NTK should change very little during training in the infinite-width limit.

At first glance, it might seem that these expectations are hardly met for our (non-infinite) experiments. Figure 1a and Figure 1c show that the relative change in the NTK during training (and also

the magnitude of the NTK) is rapidly increasing with width and remains large in magnitude for a whole range of widths of convolutional architectures. The MLP architectures do show a trend toward small changes in the NTK, yet convergence to zero is slower in the 4-Layer case than in the 2-Layer case.

However, a closer look shows that almost all of the relative change in the NTK seen in Figure 1c is explained by a simple linear re-scaling of the NTK. It should be noted that the scaling of the NTK is strongly effected by the magnitude of parameters at initialization. Within the NTK theory of Lee et al. (2017), a linear rescaling of the NTK during training corresponds simply to a change in learning rate, and so it makes more sense to measure similarity using a scale-invariant metric.

Measuring similarity between sub-sampled NTKs using the scale-invariant correlation coefficient, as in Figure 1b, is more promising. Surprisingly, we find that, as predicted in Jacot et al. (2018), the NTK changes very little (beyond a linear rescaling) for the wide ConvNet architectures. For the dense networks, the predicted trend toward small changes in the NTK also holds for most of the evaluated widths, although there is a dropoff at the end which may be an artifact of the difficulty of training these wide networks on CIFAR-10. For the Wide Residual Neural Networks, however, the general trend toward higher correlation in the wide network limit is completely reversed. The correlation coefficient decreases as network width increases, suggesting that the neural tangent kernel at initialization and after training becomes qualitatively more different as network width increases. The reversal of the correlation trend seems to be a property which emerges from the interaction of batch normalization and skip connections. Removing either of these features from the architecture leads to networks which have an almost constant correlation coefficient for a wide range of network widths, see Figure 6 in the appendix, calling for the consideration of both properties in new formulations of the NTK.

In conclusion, we see that although the NTK trends towards stability as the width of simple architectures increases, the opposite holds for the highly performant Wide ResNet architecture. Even further, neither the removal of batch normalization or the removal of skip connections fully recover the positive NTK trend. While we have hope that kernel-based theories of neural networks may yield guarantees for realistic (albeit wide) models in the future, current results do not sufficiently describe state-of-the-art architectures. Moreover, the already good behavior of models with unstable NTKs is an indicator that good optimization and generalization behaviors do not fundamentally hinge on the stability of the NTK.

## 5   RANK: DO NETWORKS WITH LOW-RANK LAYERS GENERALIZE BETTER?

State-of-the-art neural networks are highly over-parameterized, and their large number of parameters is a problem both for learning theory and for practical use. In the theoretical setting, rank has been used to tighten bounds on the generalization gap of neural networks. Generalization bounds from Harvey et al. (2017) are improved under conditions of low rank and high sparsity (Neyshabur et al., 2017) of parameter matrices, and the compressibility of low-rank matrices (and other low-dimensional structure) can be directly exploited to provide even stronger bounds (Arora et al., 2018). Further studies show a tendency of stochastic gradient methods to find low-rank solutions (Ji & Telgarsky, 2018). The tendency of SGD to find low-rank operators, in conjunction with results showing generalization bounds for low-rank operators, might suggest that the low-rank nature of these operators is important for generalization.

Langenberg et al. (2019) claim that low-rank networks, in addition to generalizing well to test data, are more robust to adversarial attacks. Theoretical and empirical results from the aforementioned paper lead the authors to make two major claims. First, the authors claim that networks which undergo adversarial training have low-rank and sparse matrices. Second, they claim that networks with low-rank and sparse parameter matrices are more robust to adversarial attacks. We find in our experiments that neither claim holds up in practical settings, including ResNet-18 models trained on CIFAR-10.

We test the generalization and robustness properties of neural networks with low-rank and high-rank operators by promoting low-rank or high-rank parameter matrices in late epochs. We employ the regularizer introduced in Sedghi et al. (2018) to create the protocols RankMin, to find low-rank parameters, and RankMax, to find high-rank parameters. RankMin involves fine-tuning a pre-trained

| Model | Training method | Clean Test Accuracy (%) | Robust (%) $\epsilon = 8/255$ | Robust (%) $\epsilon = 1/255$ |
|---|---|---|---|---|
| ResNet-18 | Natural | 94.66 | 0.00 | 31.98 |
| | RankMax | 93.66 | 0.00 | 22.01 |
| | RankMin | 94.44 | 0.00 | 31.53 |
| | Adversarial | 79.37 | 35.38 | 74.27 |
| | RankMaxAdv | 80.00 | 35.55 | 74.92 |
| | RankMinAdv | 78.34 | 33.68 | 73.19 |
| ResNet-18 w/o skips | Natural | 92.95 | 0.01 | 31.34 |
| | RankMax | 91.71 | 0.00 | 18.81 |
| | RankMin | 92.42 | 0.00 | 30.37 |
| | Adversarial | 79.57 | 35.95 | 74.88 |
| | RankMaxAdv | 79.43 | 36.45 | 74.87 |
| | RankMinAdv | 78.52 | 33.97 | 73.64 |

Table 4: Result presented here are from experiments with CIFAR-10 data and two of the architectures we studied. Robust accuracy is measured with 20-step PGD attacks with the $\epsilon$ values specified at the top of the column.

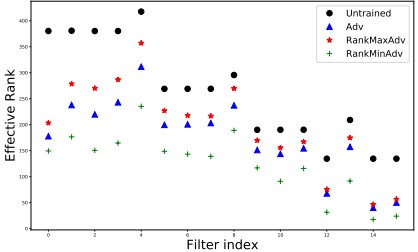

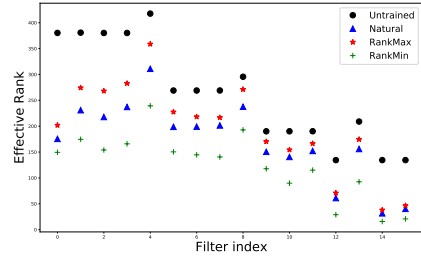

(a) Effective rank of naturally trained models.       (b) Effective rank of adversarially trained models.

Figure 2: This plot shows the effective rank of each filter for the ResNet-18 models. The filters are indexed on the $x$-axis, so moving to the right is like moving through the layers of the network. Our routines designed to manipulate the rank have exactly the desired effect as shown here.

model by replacing linear operators with their low-rank approximations, retraining, and repeating this process. Similarly, RankMax involves fine-tuning a pre-trained model by clipping singular values from the SVD of parameter matrices in order to find high-rank approximations. We are able to manipulate the rank of matrices without strongly affecting the performance of the network. We use both natural training and 7-step projected gradient descent (PGD) adversarial training routines (Madry et al., 2017). The goal of the experiment is to observe how the rank of weight matrices impacts generalization and robustness. We start by attacking naturally trained models with the standard PGD adversarial attack with $\epsilon = 8/255$. Then, we move to the adversarial training setting and test the effect of manipulating rank on generalization and on robustness.

In order to compare our results with Langenberg et al. (2019), we borrow the notion of effective rank, denoted by $r(W)$ for some matrix $W$. This continuous relaxation of rank is defined as follows. $r(W) = \frac{\|W\|_*}{\|W\|_F}$ where $\| \cdot \|_*$, $\| \cdot \|_1$, and $\| \cdot \|_F$ are the nuclear norm, the 1-norm, and the Frobenius norm, respectively. Note that the singular values of convolution operators can be found quickly with a method from Sedghi et al. (2018), and that method is used here.

In our experiments we investigate two architectures, ResNet-18 and ResNet-18 without skip connections. We train on CIFAR-10 and CIFAR-100, both naturally and adversarially. Table 4 shows that RankMin and RankMax achieve similar generalization on CIFAR-10. More importantly, when adversarially training, a setting when robustness is undeniably the goal, we see the RankMax outperforms both RankMin *and* standard adversarial training in robust accuracy. Figure 2 confirms that

these two training routines do, in fact, control effective rank. Experiments with CIFAR-100 yield similar results and are presented in Appendix A.7. It is clear that increasing rank using an analogue of rank minimizing algorithms does not harm performance. Moreover, we observe that adversarial robustness does not imply low-rank operators, nor do low-rank operators imply robustness. The findings in Ji & Telgarsky (2018) are corroborated here as the black dots in Figures 2 show that initializations are higher in rank than the trained models. Our investigation into what useful intuition in practical cases can be gained from the theoretical work on the rank of CNNs and from the claims about adversarial robustness reveals that rank plays little to no role in the performance of CNNs in the practical setting of image classification.

## 6 CONCLUSION

This work highlights the gap between deep learning theory and observations in the real-world setting. We underscore the need to carefully examine the assumptions of theory and to move past the study of toy models, such as deep linear networks or single-layer MLPs, whose traits do not describe those of the practical realm. First, we show that realistic neural networks on realistic learning problems contain suboptimal local minima. Second, we show that low-norm parameters may not be optimal for neural networks, and in fact, biasing parameters to a non-zero norm during training improves performance on several popular datasets and a wide range of networks. Third, we show that the wide-network trends in the neural tangent kernel do not hold for ResNets and that the interaction between skip connections and batch normalization play a large role. Finally, we show that low-rank linear operators and robustness are not correlated, especially for adversarially trained models.

## ACKNOWLEDGMENTS

This work was supported by the AFOSR MURI Program, the National Science Foundation DMS directorate, and also the DARPA YFA and L2M programs. Additional funding was provided by the Sloan Foundation.

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

## A APPENDIX

### A.1 PROOF OF LEMMA 1

**Lemma 1.** *Consider a family of L-layer multilayer perceptrons with ReLU activations $\{F_\phi : \mathbb{R}^m \to \mathbb{R}^n\}$ and let $s = \min_i n_i$ be the minimum layer width. Then this family has rank-$s$ affine expression.*

*Proof.* Let $G$ be a rank-$s$ affine function, and $\Omega \subset \mathbb{R}^m$ be a finite set. Let $G(\mathbf{x}) = A\mathbf{x} + \mathbf{b}$ with $A = U\Sigma V$ being the singular value decomposition of $A$ with $U \in \mathbb{R}^{n \times s}$ and $V \in \mathbb{R}^{s \times m}$.

We define

$$A_1 = \begin{bmatrix} \Sigma V \\ \mathbf{0} \end{bmatrix}$$

where $\mathbf{0}$ is a (possibly void) $(n_1 - s) \times m$ matrix of all zeros, and $b_1 = c\mathbf{1}$ for $c = \max_{\mathbf{x}_i \in \Omega, 1 \le j \le n_1} |(A_1 \mathbf{x}_i)_j| + 1$ and $\mathbf{1} \in \mathbb{R}^{n_1}$ being a vector of all ones. We further choose $A_l \in \mathbb{R}^{n_l \times n_{l-1}}$ to have an $s \times s$ identity matrix in the upper left, and fill all other entries with zeros. This choice is possible since $n_l \ge s$ for all $l$. We define $\mathbf{b}_l = \begin{bmatrix} \mathbf{0} & c\,\mathbf{1} \end{bmatrix}^T \in \mathbb{R}^{n_l}$ where $\mathbf{0} \in \mathbb{R}^{1 \times s}$ is a vector of all zeros and $\mathbf{1} \in \mathbb{R}^{1 \times (n_l - s)}$ is a (possibly void) vector of all ones.

Finally, we choose $A_L = \begin{bmatrix} U & \mathbf{0} \end{bmatrix}$, where now $\mathbf{0}$ is a (possibly void) $n \times (n_{L-1} - s)$ matrix of all zeros, and $\mathbf{b}_L = -cA_L\mathbf{1} + \mathbf{b}$ for $\mathbf{1} \in \mathbb{R}^{n_{L-1}}$ being a vector of all ones.

Then one readily checks that $F_\phi(\mathbf{x}) = G(\mathbf{x})$ holds for all $x \in \Omega$. Note that all entries of all activations are greater or equal to $c > 0$, such that no ReLU ever maps an entry to zero. □

### A.2 PROOF OF THEOREM 1

**Theorem 1.** *Consider a training set, $\{(\mathbf{x}_i, y_i)\}_{i=1}^N$, a family $\{F_\phi\}$ of MLPs with $s = \min_i n_i$ being the smallest width. Consider the training of a rank-$s$ linear classifier $G_{A,\mathbf{b}}$, i.e.,*

$$\min_{A,\mathbf{b}} \mathcal{L}(G_{A,\mathbf{b}}; \{(\mathbf{x}_i, y_i)\}_{i=1}^N), \qquad \text{subject to } rank(A) \le s, \tag{5}$$

*for any continuous loss function $\mathcal{L}$. Then for each local minimum, $(A', \mathbf{b}')$, of the above training problem, there exists a local minimum, $\phi'$, of $\mathcal{L}(F_\phi; \{(\mathbf{x}_i, y_i)\}_{i=1}^N)$ with the property that $F_{\phi'}(\mathbf{x}_i) = G_{A', \mathbf{b}'}(\mathbf{x}_i)$ for $i = 1, 2, ..., N$.*

*Proof.* Based on the definition of a local minimium, there exists an open ball $D$ around $(A', \mathbf{b}')$ such that

$$\mathcal{L}(G_{A', \mathbf{b}'}; \{(\mathbf{x}_i, y_i)\}_{i=1}^N) \leq \mathcal{L}(G_{A, \mathbf{b}}; \{(\mathbf{x}_i, y_i)\}_{i=1}^N) \quad \forall (A, \mathbf{b}) \in D \text{ with rank}(A) \leq s. \quad (6)$$

First, we use the same construction as in the proof of Lemma 1 to find a function $F_{\phi'}$ with $F_{\phi'}(\mathbf{x}_i) = G_{A', \mathbf{b}'}(\mathbf{x}_i)$ for all training example $\mathbf{x}_i$. Because the mapping $\phi \mapsto F_\phi(\mathbf{x}_i)$ is continuous (not only for the entire network $F$ but also for all subnetworks), and because all activations of $F_{\phi'}$ are greater or equal to $c > 0$, there exists an open ball $B(\phi', \delta_1)$ around $\phi'$ such that the activations of $F_\phi$ remain positive for all $\mathbf{x}_i$ and all $\phi \in B(\phi', \delta_1)$.

Consequently, the restriction of $F_\phi$ to the training set remains affine linear for $\phi \in B(\phi', \delta_1)$. In other words, for any $\phi \in B(\phi', \delta_1)$ we can write

$$F_\phi(\mathbf{x}_i) = A(\phi)\mathbf{x}_i + \mathbf{b}(\phi) \qquad \forall \mathbf{x}_i,$$

by defining $A(\phi) = A_L A_{L-1} \ldots A_1$ and $\mathbf{b}(\phi) = \sum_{l=1}^L A_L A_{L-1} \ldots A_{l+1} \mathbf{b}_l$. Note that due to $s = \min_i n_i$, the resulting $A(\phi)$ satisfies rank$(A(\phi)) \leq s$.

After restricting $\phi$ to an open ball $B(\phi', \delta_2)$, for $\delta_2 \leq \delta_1$ sufficiently small, the above $(A(\phi), \mathbf{b}(\phi))$ satisfy $(A(\phi), \mathbf{b}(\phi)) \in D$ for all $\phi \in B(\phi', \delta_2)$. On this set, we, however, already know that the loss can only be greater or equal to $\mathcal{L}(F_{\phi'}; \{(\mathbf{x}_i, y_i)\}_{i=1}^N)$ due to equation 6. Thus, $\phi'$ is a local minimum of the underlying loss function. $\square$

### A.3 ADDITIONAL COMMENTS REGARDING THEOREM 1

Note that our theoretical and experimental results do not contradict theoretical guarantees for deep linear networks (Kawaguchi, 2016; Laurent & Brecht, 2018) which show that all local minima are global. A deep linear network with $s = \min(n, m)$ is equivalent to a linear classifier, and in this case, the local minima constructed by Theorem 1 are global. However, this observation shows that Theorem 1 characterizes the gap between deep linear and deep nonlinear networks; the global minima predicted by linear network theories are inherited as (usually suboptimal) local minima when ReLU's are added. Thus, linear networks do not accurately describe the distribution of minima in non-linear networks.

### A.4 ADDITIONAL RESULTS FOR SUBOPTIMAL LOCAL OPTIMA

Table 5 shows more experiments. As above in the previous experiment, we use gradient descent to train a full ResNet-18 architecture on CIFAR-10 until convergence from different initializations. We find that essentially the same results appear for the deeper architecture, initializing with very high bias leads to highly non-optimal solutions. In this case even solutions that are equally bad as a zero-norm initialization.

Further results on CIFAR-100 are shown in Tables 6 and 7. These experiments with MLP and ResNet-18 show the same trends as explained above, thus confirming that the results are not specific to the CIFAR-10 dataset.

### A.5 DETAILS CONCERNING LOW-NORM REGULARIZATION EXPERIMENTS

Our experiments comparing regularizers all run for 300 epochs with an initial learning rate of 0.1 and decreases by a factor of 10 at epochs 100, 175, 225, and 275. We use the SGD optimizer with momentum 0.9.

We also tried negative weight decay coefficients, which leads to ResNet-18 CIFAR-10 performance above 90% while blowing up parameter norm, but this performance is still suboptimal and is not informative concerning the optimality of minimum norm solutions. One might wonder if high norm-bias coefficients lead to even lower parameter norm than low weight decay coefficients. This

Table 5: Local minima for ResNet-18 and CIFAR-10 generated via initialization and trained by vanilla gradient descent, showing loss, euclidean norm of the gradient vector.

| | At Initialization | | After training | |
|---|---|---|---|---|
| Init. Type | Loss | Grad. | Loss | Grad. |
| Default | 2.30312 | 0.05000 | 0.00014 | 0.01410 |
| Zero | 2.30258 | 0.00025 | 2.30259 | 0.00013 |
| Bias+20 | 12.95754 | 590.12170 | 2.30658 | 0.00004 |
| Bias $\in \mathcal{U}(-10, 10)$ | 12.96790 | 214.68600 | 2.30260 | 0.00123 |
| Bias $\in \mathcal{U}(-50, 50)$ | 84.67800 | 1190.23500 | 2.30260 | 0.00702 |

Table 6: Local minima for ResNet-18 and CIFAR-100 generated via initialization and trained by vanilla gradient descent, showing loss, euclidean norm of the gradient vector

| | At Initialization | | After training | |
|---|---|---|---|---|
| Init. Type | Loss | Grad. | Loss | Grad. |
| Default | 4.60591 | 0.02346 | 0.00030 | 0.00466 |
| Zero | 4.60517 | 0.00019 | 4.60517 | 0.00003 |
| Bias+20 | 34.37053 | 655.51569 | 4.60517 | 0.00015 |
| Bias $\in \mathcal{U}(-100, 100)$ | 178.74391 | 2615.72534 | 4.60517 | 0.00003 |

Table 7: Local minima for MLP and CIFAR-100 generated via initialization and trained by vanilla gradient descent, showing loss, euclidean norm of the gradient vector.

| | At Initialization | | After training | |
|---|---|---|---|---|
| Init. Type | Loss | Grad. | Loss | Grad. |
| Default | 4.60670 | 0.16154 | 0.02579 | 0.01482 |
| Zero | 4.60517 | 0.00019 | 4.60517 | 0.00011 |
| Bias+10 | 15.77286 | 359.65710 | 4.60517 | 0.00079 |
| Bias $\in \mathcal{U}(-5, 5)$ | 8.69149 | 63.59983 | 2.15917 | 0.09718 |
| Bias $\in \mathcal{U}(-10, 10)$ | 13.02693 | 158.78347 | 2.58368 | 0.09233 |

question may not be meaningful in the case of networks with batch normalization. In the case of ResNet-20 with Fixup, which does not contain running mean and standard deviation, the average parameter $\ell_2$ norm after training with weight decay is $24.51$ while that of models trained with norm-bias is $31.62$. Below, we perform the same tests on CIFAR-100, a substantially more difficult dataset. Weight decay coefficients are chosen to be ones used in the original paper for the corresponding architecture. Norm-bias coefficient/$\mu^2$ is chosen to be $8100/0.005$, $7500/0.001$, and $2000/0.0005$ for ResNet-18, DenseNet-40, and ResNet-20 with Fixup, respectively, using the same heuristic as described in the main body.

Table 8: ResNet-18, DenseNet-40, and ResNet-20 with Fixup initialization trained on normalized CIFAR-100 data with various regularizers. Numerical entries are given by $\overline{m}(\pm s)$, where $\overline{m}$ is the average accuracy over 10 runs, and $s$ represents standard error.

| Model | No weight decay (%) | Weight decay (%) | Norm-bias (%) |
|---|---|---|---|
| ResNet | 71.73 ($\pm$0.25) | 74.66 ($\pm$0.17) | **75.90** ($\pm$0.16) |
| DenseNet | 65.61 ($\pm$0.33) | 68.98 ($\pm$0.25) | **69.24** ($\pm$0.11) |
| ResNet Fixup | 1.000 ($\pm$0.00) | 65.08 ($\pm$0.30) | **65.58** ($\pm$0.17) |

## A.6 DETAILS ON THE NEURAL TANGENT KERNEL EXPERIMENT

For further reference, we include details on the NTK sampling during training epochs in Figure 3. We see that the parameter norm (Right) behaves normally (all of these experiments are trained with a standard weight decay parameter of $0.0005$), yet the NTK norm (Left) rapidly increases. Most of this increase, however is scaling of the kernel, as the correlation plot (Middle) is much less drastic. We do see that most change happens in the very first epochs of training, whereas the kernel only changes slowly later on.

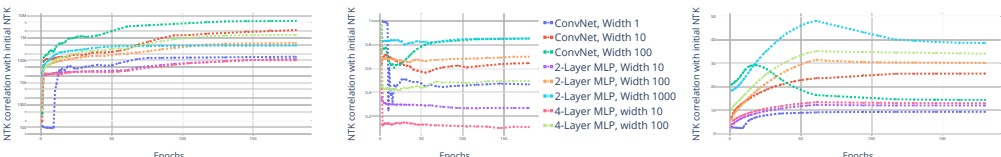

Figure 3: Plotting the evolution of NTK parameters during training epochs. Left: Norm of the NTK Tensor, Middle: Correlation of current NTK iterate versus initial NTK. Right: Reference plot of the network parameter norms.

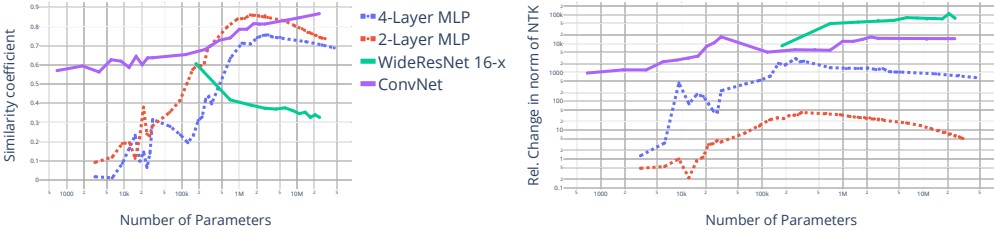

Figure 4: The similarity coefficient of the neural tangent kernel after training with its initialization. We expect this coefficient to converge toward 1 in the infinite-width limit for multi-layer networks. Also shown is the direct relative difference of the NTK norms, which behaves similarly to the normalized direct difference from figure 1.

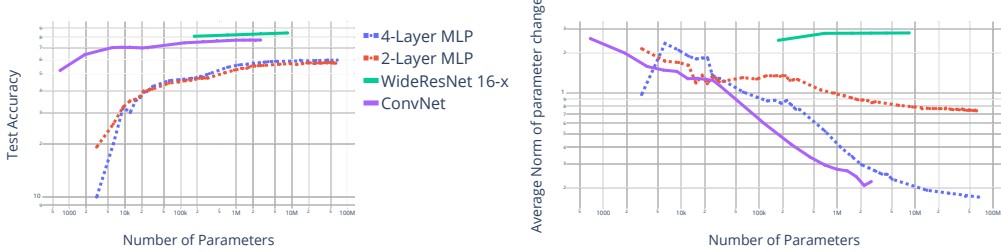

Figure 5: For reference we record the test accuracy of all models from 1 in the left plot and the relative change in parameters in the right plot.

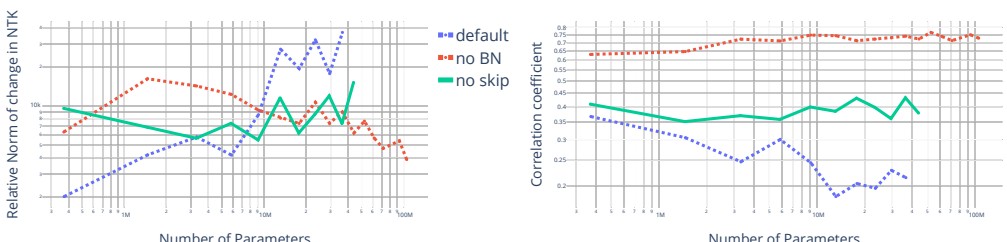

Figure 6: The correlation coefficient of the neural tangent kernel after training with its initialization for different WideResNet variants - namely WideResNet without batch normalizations and WideResNet without skip connections. We interestingly find that removing either of both properties, which are widely regarding as beneficial for neural network training, stabilizes the trend seen in the default WideResNet. However both variants hardly converge toward 1, even when sampling very wide ResNets.

## A.7 DETAILS ON RANKMIN AND RANKMAX

We employ routines to promote both low-rank and high-rank parameter matrices. We do this by computing approximations to the linear operators at each layer. Since convolutional layers are linear operations, we know that there is a matrix whose dimensions are the number of parameters in the input to the convolution and the number of parameters in the output of the convolution. In order to compute low-rank approximations of these operators, one could write down the matrix corresponding to the convolution, and then compute a low-rank approximation using a singular value decomposition (SVD). In order to make this problem computationally tractable we used the method for computing singular values of convolution operators derived in Sedghi et al. (2018). We were then able to do low-rank approximation in the classical sense, by setting each singular value below some threshold to zero. In order to compute high-rank operators, we clipped the singular values so that when mulitplying the SVD factors, we set each singular value to be equal to the minimum of some chosen constant and the true singular value. It is important to note here that these approximations to the convolutional layers, when done naively, can return convolutions with larger filters. To be precise, an $n \times n$ filter will map to a $k \times k$ filter through our rank modifications, where $k \geq n$. We follow the method in Sedghi et al. (2018), where these filters are pruned back down by only using $n \times n$ entries in the output.

When naturally training ResNet-18 and Skipless ResNet-18 models, we train with a batch size of 128 for 200 epochs with the learning rate initiated to 0.01 and decreasing by a factor of 10 at epochs 100, 150, 175, and 190 (for both CIFAR-10 and CIFAR-100). When adversarially training these two models on CIFAR-10 data, we use the same hyperparameters. However, in order to adversarially train on CIFAR-100, we train ResNet-18 with a batch size of 256 for 300 epochs with an initial learning rate of 0.1 and a decrease by a factor of 10 at epochs 200 and 250. For adversarially training Skipless ResNet-18 on CIFAR-100, we use a batch size of 256 for 350 epochs with an

initial learning rate of 0.1 and a decrease by a factor of 10 at epochs 200, 250, and 300. Adversarial training is done with an $\ell_\infty$ 7-step PGD attack with a step size of $2/255$, and $\epsilon = 8/255$. For all of the training described above we augment the data with random crops and horizontal flips.

During 15 additional epochs of training we manipulate the rank as follows. RankMin and RankMax protocols are employed periodically in the last 15 epochs taking care to make sure that the loss remains small. For these last epochs, the learning rate starts at 0.001 and decreases by a factor of 10 after the third and fifth epochs of the final 15 epochs. As shown in Table 10, we test the accuracy of each model on clean test data from the corresponding dataset, as well as on adversarial examples generated with 20-step PGD with $\epsilon = 8/255$ (with step size equal to $2/255$) and $\epsilon = 1/255$ (with step size equal to $.25/255$).

When training multi-layer perceptrons on CIFAR-10, we train for 100 epochs with learning rate initialized to 0.01 and decreasing by a factor of 10 at epochs 60, 80 and 90. Then, we train the network for 8 additional epochs, during which RankMin and RankMax networks undergo rank manipulation.

Table 9: Results from rank experiments with a multi-layer perceptron and CIFAR-10.

MLP and CIFAR-10

| Training method | Training Accuracy (%) | Clean Accuracy (%) | Robust (%) $\epsilon = 8/255$ | Robust (%) $\epsilon = 1/255$ |
|---|---|---|---|---|
| Naturally Trained | 100.00 | 58.79 | 3.76 | 28.94 |
| RankMax | 99.97 | 58.19 | 3.72 | 26.63 |
| RankMin | 100.00 | 58.06 | 3.76 | 28.48 |

Table 10: Results from rank experiments on CIFAR-100. Robust accuracy is measured with 20-step PGD attacks with the $\epsilon$ values specified at the top of the column.

ResNet-18 and CIFAR-100

| Training method | Training Accuracy (%) | Clean Accuracy (%) | Robust (%) $\epsilon = 8/255$ | Robust (%) $\epsilon = 1/255$ |
|---|---|---|---|---|
| Naturally Trained | 99.97 | 73.08 | 0.00 | 17.5 |
| RankMax | 99.90 | 72.67 | 0.00 | 16.95 |
| RankMin | 99.92 | 72.57 | 0.00 | 17.63 |
| Adversarially Trained | 99.92 | 50.88 | 17.81 | 45.99 |
| RankMaxAdv | 99.73 | 51.04 | 16.80 | 45.74 |
| RankMinAdv | 99.91 | 50.22 | 16.64 | 45.03 |

ResNet-18 w/o skip connections and CIFAR-100

| Training method | Training Accuracy (%) | Clean Accuracy (%) | Robust (%) $\epsilon = 8/255$ | Robust (%) $\epsilon = 1/255$ |
|---|---|---|---|---|
| Naturally Trained | 99.96 | 72.13 | 0.01 | 13.7 |
| RankMax | 99.82 | 71.35 | 0.04 | 11.74 |
| RankMin | 99.90 | 71.28 | 0.00 | 13.53 |
| Adversarially Trained | 99.92 | 50.47 | 17.62 | 45.18 |
| RankMaxAdv | 99.90 | 50.93 | 17.72 | 45.78 |
| RankMinAdv | 99.91 | 49.37 | 16.77 | 44.41 |

