# OpenReview forum: "Truth or backpropaganda? An empirical investigation of deep learning theory"
_ICLR.cc/2020/Conference — Accept (Spotlight)_

### Official Review · AnonReviewer3 · 2019-10-23
**Official Blind Review #3**

**Rating:** 8

**Review:**

The authors seek to challenge some presumptions about training deep neural networks, such as the robustness of low rank linear layers and the existence of suboptimal local minima. They provide analytical insight as well as a few experiments.

I give this paper an accept. They analytically explore four relevant topics of deep learning, and provide experimental insight. In particular, they provide solid analytical reasoning behind their claims that suboptimal local minima exist and that their lack of prevalence is due to improvements in other aspects of deep networks, such as initialization and optimizers. In addition, they present a norm-bias regularizer generalization that consistently increases accuracy. I am especially pleased with this, as the results are averaged over several runs (a practice that seems to be not so widespread these days).

If I were to have one thing on my wish list for this paper, it would be the small issue of having some multiple experiment version of the local minima experiments (I understand why it is not all that necessary for the rank and stability experiments).

Nevertheless, I think this paper gives useful insight as to the behavior of deep neural networks that can help advance the field on a foundational level.

**Experience Assessment:**

I have read many papers in this area.

**Review Assessment: Checking Correctness Of Derivations And Theory:**

I assessed the sensibility of the derivations and theory.

**Review Assessment: Checking Correctness Of Experiments:**

I assessed the sensibility of the experiments.

**Review Assessment: Thoroughness In Paper Reading:**

I read the paper at least twice and used my best judgement in assessing the paper.

---

> ### Author Response · Authors · 2019-11-14
> **Reply to reviewer #3**
>
> We appreciate the positive feedback, and we thank the reviewer for the thoughtful comments. We have added results from further suboptimal minima experiments to the appendix.

---

### Official Review · AnonReviewer1 · 2019-10-23
**Official Blind Review #1**

**Rating:** 6

**Review:**

The authors look at empirical properties of deep neural networks and discuss their connection to past theoretical work on the following issues:

* Local minima: they give an example of setting where bad local minima (far from the global minimum) are obtained. More specifically, they show such minima can be obtained by initializing with large random biases for MLPs with ReLU activation. They also provide a theoretical result that can be used to find a small set of such minima. I believe this is a useful incremental step towards a better understanding of local minima in deep learning, although it is not clear how many practical implications this has. One question that would ideally be answered is: in practical settings, to what degree does bad initialization cause bad performance specifically due to bad minima? (as opposed to, say, slow convergence or bad generalization performance).

* Weight decay: the authors penalize the size of the norm of the weights as it diverges from a constant, as opposed to when it diverges from 0 as is normally done for weight decay. They show that this works as well or better than normal weight decay in a number of settings. This seem to put into question the belief sometimes held that solutions with smaller norms will generalize better.

* Kernel theory: the authors try to reproduce some of the empirical properties predicted in the Neural Tangent Kernel paper (Jacot et al., 2018) in particular by using more realistic architectures. The results, however, do not appear very conclusive. This might be the weakest part of the paper, as it is hard to draw anything conclusive from their empirical results.

* Rank: The authors challenge the common belief that low rank provides better generalization and more robustness towards adversarial attacks. When enforcing a low or high rank weight matrices during training on ResNet-18 trained on CIFAR-10, the two settings have similar performance and are similarly robust to adversarial attacks, showing at least one counter example.

I think overall this is a useful although somewhat incremental paper, that makes progress in the understanding of the behavior of neural networks in practice, and can help guide further theoretical work and the  development of new and improved training techniques and initialization regimes for deep learning.

Other comments/notes:
* minor: the order of the last 2 sub topics covered (rank and NTK) is flipped in the introduction, compared to the abstract and the order of the chapters
* in the table confidence intervals are given, it would be nice to have more details on how they are computed, (e.g. +- 1.96 * std error)
* how is the constant \mu in the norm-bias chosen?

**Experience Assessment:**

I have read many papers in this area.

**Review Assessment: Checking Correctness Of Derivations And Theory:**

I assessed the sensibility of the derivations and theory.

**Review Assessment: Checking Correctness Of Experiments:**

I assessed the sensibility of the experiments.

**Review Assessment: Thoroughness In Paper Reading:**

I read the paper at least twice and used my best judgement in assessing the paper.

---

> ### Author Response · Authors · 2019-11-14
> **Reply to reviewer #1**
>
> Thank you for your thoughtful input on our work. We address your comments in order:
> * Our work here is focused on finding suboptimal minima, and we show that certain poor initializations motivated by theory can lead to this.  We agree that suboptimal local minima which arise from bad initializations in standard practice would be interesting to study in future work.
> * We have changed the conclusion of the NTK section to more clearly discuss and conceptualize our findings, and we have added additional plots.
> * The order of the topics has been fixed, thank you for bringing this to our attention.
> * We have added details regarding the confidence intervals.
> * The constant \mu is chosen heuristically by studying the norm of parameter vectors that result from standard weight decay, and setting \mu to be higher to make sure that networks trained with norm-bias indeed have a higher norm than those trained with weight decay. This explanation is now included in the section on weight norms.

---

### Official Review · AnonReviewer2 · 2019-10-27
**Official Blind Review #2**

**Rating:** 8

**Review:**

In this paper, the authors seek to examine carefully some assumptions investigated in the theory of deep neural networks. The paper attempts to answer the following theoretical assumptions: the existence of local minima in loss landscapes, the relevance of weight decay with small L2-norm solutions, the connection between deep neural networks to kernel-based learning theory, and the generalization ability of networks with low-rank layers.

We think that this work is timely and of significant interest, since theoretical work on deep learning has made significant progress in recent years.

Since this paper seeks to provide an empirical study on the assumptions in deep learning theory, we think that the results are somehow weak as the paper is missing extensive analysis, using several well-known datasets and several deep architectures and settings. For example, only the CIFAR-10 dataset is considered in the paper, and it is not clear whether the obtained results will generalize to other datasets. This also goes to the neural network architecture, as only MLP is considered to answer the assumption about the existence of suboptimal minima, while only ResNet is considered to study the generalization abilities with low-rank layers. We think that this is not enough for a paper that tries to provide an empirical study.

-------
Reply to rebuttal

We thank the authors for taking into consideration our previous comments and suggestions, including going beyond MLP and adding experiments on other datasets. For this reason, we have increased the rating from "Weak Accept" to "Accept".

**Experience Assessment:**

I have read many papers in this area.

**Review Assessment: Checking Correctness Of Derivations And Theory:**

I assessed the sensibility of the derivations and theory.

**Review Assessment: Checking Correctness Of Experiments:**

I assessed the sensibility of the experiments.

**Review Assessment: Thoroughness In Paper Reading:**

I made a quick assessment of this paper.

---

> ### Author Response · Authors · 2019-11-14
> **Reply to reviewer #2**
>
> We thank the reviewer for the time and effort spent on our paper. We agree about the note concerning the breadth of our experiments and have made the following additions to the paper.
> * Experiments have been run on CIFAR-100 data and the results, which agree with our previous findings, are in the appendix.
> * Our study of suboptimal minima had included experiments with ResNet-18. Results are in the appendix. As mentioned above, we have since added these experiments on CIFAR-100 for diversity of data sets.
> * The section on rank has been updated to reflect further experiments with new architectures. Specifically, we tested ResNet-18 without skip connections and MLP. See the updated appendix for full details and results.

---

### Public Comment · ~Pedro_Tabacof1 · 2019-10-03
**Bayesian Neural Network Ensembles connection**

The work Bayesian Neural Network Ensembles by Pearce et al (https://arxiv.org/abs/1811.12188) proposes to use an ensemble of neural networks that are each trained with L2 regularization from a normal distribution sample. They show this is form of approximate Bayesian inference.

That idea is somewhat similar to the "norm-bias" regularizer proposed in this paper, with the difference that the weights attracted to a normal distribution sample rather than a fixed value.

I just wanted to point out this connection, which may be relevant to explain why norm-bias works.

---

> ### Author Response · Authors · 2019-10-04
> **Interesting connection**
>
> Hi Pedro,
> Thank you for pointing out this paper.  We agree that the relationship between explicit regularizers, like weight decay and norm-bias, during training and Bayesian priors at inference may be an interesting direction for future work.

---

### Public Comment · ~Alex_Matthew_Lamb1 · 2019-10-05
**Comments about Section 5 on Rank**


1.  As far as I can tell, this section discusses the issue of the weight matrices in a deep network being low-rank.  However the discussion in the literature focuses on both the rank of the weight matrices as well as the rank of the hidden states.

It's not clear to me how these issues are related to each other, especially in non-linear networks.  Do you think the results in your paper also have some relevance for the study of low-rank hidden states or would you consider it to be separate issue?

2.  Do you think it's worth analyzing the resnet and non-resnet cases separately here?  If I think about a linear neural network, the residual network variant will always amount to a full-rank affine transformation on each layer as a result of the skip connection, even if the applied weight matrix W is low-rank.

---

> ### Author Response · Authors · 2019-10-10
> **Rank in residual networks**
>
> Thank you for the insightful comments:
> 1)  We only consider low-rank linear operators since this topic is studied in the generalization and robustness works we discuss such as Neyshabur et al. (2017) and Langenberg et al. (2019).  However, we agree that low-rank hidden states may also be an interesting topic for empirical work.
> 2)  The rank of linear operators in ResNets indeed contributes differently to the behavior of the network than the rank of linear operators in MLPs.  In the case of linear ResNets, for example, if the applied weight matrix is the negative identity, then after a skip connection, the combined layer would be rank-0.  In fact, the combined layer which includes a skip connection may be a low or high rank affine transformation.  However, in the non-linear case, it is not clear what the analogous rank measurements would be since there are nonlinearities between affine transformations and skip connections, and thus, we cannot collapse layers and skip connections into one combined affine transformation.
> We chose ResNet-18 in order to determine if intuitions developed by theory transfer to a realistic architecture.  Even so, we have also tested these claims in the context of MLPs and found that the same results hold as do for ResNets in the case of generalization, and more notably, a naturally trained MLP with RankMax achieves higher robust accuracy than the same MLP trained with RankMin.  For example, an MLP with RankMin achieved 49.94% robust accuracy on CIFAR-10 against the small-radius PGD attack from the paper, while the same MLP with RankMax achieved 51.45% robust accuracy.  We may include these MLP results in the next version of our paper if there is interest.

---

> > ### Public Comment · ~Amartya_Sanyal2 · 2019-10-21
> > **Low Rank Representations**
> >
> > 1. Given the discussion about low rank hidden states, I thought I would point out our work on low rank representations and its effect on adversarial robustness.
> >  https://arxiv.org/abs/1804.07090
> >
> > 2. While Linear operators can make the pre-activations low rank, (and the skip connection may or may not increase its rank), the non-linear activation function often increases it to give high rank hidden states (eg. Appendix A in the paper above).

---

> > > ### Author Response · Authors · 2019-10-23
> > > **Interesting research direction**
> > >
> > > Thank you for letting us know about your paper.  The phenomenon of low-rank hidden states is an interesting direction for research.

---

### Public Comment · ~Chulhee_Yun1 · 2019-10-19
**Relevant work on the existence of bad local minima**

Dear authors,

I enjoyed reading your submission. Thanks for the interesting paper!

After reading the paper, I wanted to bring to your attention a paper of ours on the existence of bad local minima that seems quite relevant:
[1] Small nonlinearities in activation functions create bad local minima in neural networks, ICLR 2019, https://openreview.net/forum?id=rke_YiRct7

In particular, Theorem 1 of [1] constructs local minima of neural networks whose predictions perform just as well as the linear predictor, and shows that for general datasets that these local minima are not globally optimal. As far as I understand, the key idea of the proof of Theorem 1 in this submission looks very similar to [1]: pushing the bias high enough so that the network becomes linear. In my opinion, the theoretical results in this submission and [1] are highly relevant, so it would be very helpful if the authors could compare them in the paper.

I’d also like to note that Theorem 1 of [1] also implies that even with slightest nonlinearity (slope 1+\epsilon on positive side and slope 1 on negative side) and for general datasets, there exist bad local minima. Furthermore, I believe the assumptions in [1] are milder than the other previous results cited in Section 2 of this submission.

Overall, I believe [1] is highly relevant to this submission. Thus, we would appreciate it if the authors could cite our paper as well as contextualize their results with ours. We hope that the authors will be able to bring out the differences and potential subtleties, if any.

Thank you!
Charlie Yun

---

> ### Author Response · Authors · 2019-10-21
> **Interesting previous work**
>
> Hello Charlie,
>
> Thank you for bringing your work to our attention.  We agree that it is highly relevant, and we are eager to discuss and contextualize these results in the next version of this submission. We agree that previous work on the existence of local minima was limited in comparison to [1] and find that your work bridges the gap between these and our work.
>
> In terms of differences, Theorem 1 from [1], to our understanding, applies to networks with a single hidden layer and squared error, whereas Theorem 1 in our work applies to networks of arbitrary depth and any continuous loss function. Furthermore, we do not assume that all data points are unique and that output is one-dimensional.
>
> Aside from these more technical terms, we think it is crucial to note that even if the data can be fitted with a linear classifier of dimension m, our work shows that any network with a smaller width n still contains spurious local minima, corresponding to linear classifiers with rank <= n. What we further find interesting is that our result can be recursively extended to local minima at which a network behaves like a shallower subnetwork on the training data.  This extension may not follow directly from [1] since outputs are univariate.  Our proof also applies to networks with convolutional layers since they can form the identity necessary for our construction.
>
> We further like the idea of generalizing to other activation functions.   We chose ReLUs for simplicity and their wide use, but any activation functions which are affine-linear with nonzero slope on some open interval are equally suitable under our proof technique.  Such a corollary inspired by your variant would be a good fit for the next version.
>
> Best Regards,
> The Authors
>
> [1] Small nonlinearities in activation functions create bad local minima in neural networks, ICLR 2019

---

### Decision · Program_Chairs · 2019-12-19

**Decision:**

Accept (Spotlight)

**Comment:**

The authors take a closer look at widely held beliefs about neural networks. Using a mix of analysis and experiment, they shed some light on the ways these assumptions break down. The paper contributes to our understanding of various phenomena and their connection to generalization, and should be a useful paper for theoreticians searching for predictive theories.